# Prognostic Impact of Hybrid Comprehensive Telerehabilitation Regarding Diastolic Dysfunction in Patients with Heart Failure with Reduced Ejection Fraction—Subanalysis of the TELEREH-HF Randomized Clinical Trial

**DOI:** 10.3390/jcm11071844

**Published:** 2022-03-26

**Authors:** Robert Irzmański, Renata Glowczynska, Maciej Banach, Dominika Szalewska, Ryszard Piotrowicz, Ilona Kowalik, Michael J. Pencina, Wojciech Zareba, Piotr Orzechowski, Slawomir Pluta, Zbigniew Kalarus, Grzegorz Opolski, Ewa Piotrowicz

**Affiliations:** 1Department of Internal Medicine and Cardiac Rehabilitation, Medical University of Łódź, 90-647 Lodz, Poland; robert.irzmanski@umed.lodz.pl; 21st Chair and Department of Cardiology, Medical University of Warsaw, 02-097 Warsaw, Poland; grzegorz.opolski@wum.edu.pl; 3Department of Hypertension, Medical University of Łódź, 90-647 Lodz, Poland; maciej.banach@umed.lodz.pl; 4Clinic of Rehabilitation Medicine, Faculty of Health Sciences, Medical University of Gdańsk, 80-210 Gdańsk, Poland; dominika.szalewska@gumed.edu.pl; 5National Institute of Cardiology, 04-628 Warsaw, Poland; rpiotrowicz@ikard.pl (R.P.); ikowalik@ikard.pl (I.K.); 6Warsaw Academy of Medical Rehabilitation, 01-234 Warsaw, Poland; 7The Department of Biostatistics and Bioinformatics, Duke University School of Medicine, Durham, NC 27710, USA; michal.pencina@duke.edu; 8Cardiology Unit of the Department of Medicine, University of Rochester Medical Center, Rochester, NY 14642, USA; wojciech_zareba@urmc.rochester.edu; 9Telecardiology Center, National Institute of Cardiology, 04-628 Warsaw, Poland; porzechowski@ikard.pl (P.O.); epiotrowicz@ikard.pl (E.P.); 10Department of Cardiology, Congenital Heart Diseases and Electrotherapy, Silesian Center for Heart Diseases, Silesian Medical University, 41-800 Zabrze, Poland; spluta77@gmail.com (S.P.); zbigniewkalarus@kalmet.com.pl (Z.K.)

**Keywords:** hybrid telerehabilitation, heart failure with reduced ejection fraction, diastolic function

## Abstract

Aims: The objective of the study was to evaluate the effects of individually prescribed hybrid comprehensive telerehabilitation (HCTR) implemented at patients’ homes on left ventricular (LV) diastolic function in heart failure (HF) patients. Methods and results: The Telerehabilitation in Heart Failure Patients trial (TELEREH-HF) is a multicenter, prospective, randomized (1:1), open-label, parallel-group, controlled trial involving HF patients assigned either to HCTR involving a remotely monitored home training program in conjunction with usual care (HCTR group) or usual care only (UC group). The patient in the HCTR group underwent a 9-week HCTR program consisting of two stages: an initial stage (1 week) conducted in hospital and the subsequent stage (eight weeks) of home-based HCTR five times weekly. Due to difficulties of proper assessment and differences in the evaluation of diastolic function in patients with atrial fibrillation, we included in our subanalysis only patients with sinus rhythm. Depending on the grade of diastolic dysfunction, patients were assigned to subgroups with mild diastolic (MDD) or severe diastolic dysfunction (SDD), both in HCTR (HCTR-MDD and HCTR-SDD) and UC groups (UC-MDD and UC-SDD). Changes from baseline to 9 weeks in echocardiographic parameters were seen only in A velocities in HCTR-MDD vs. UC-MDD; no significant shifts between groups of different diastolic dysfunction grades were observed after HCTR. All-cause mortality was higher in UC-SDD vs. UC-MDD with no difference between HCTR-SDD and HCTR-MDD. Higher probability of HF hospitalization was observed in HCTR-SDD than HCTR-MDD and in UC-SDD than UC-MDD. No differences in the probability of cardiovascular mortality and hospitalization were found. Conclusions: HCTR did not influence diastolic function in HF patients in a significant manner. The grade of diastolic dysfunction had an impact on mortality only in the UC group and HF hospitalization over a 12–24-month follow-up in HCTR and UC groups.

## 1. Introduction

Heart failure (HF) is a major challenge in modern healthcare and is increasing with the aging of the population. The pathophysiology in HF is determined by altered cardiac output, reduced cardiac contractility, myocardial stiffness, increased filling pressure of LV and diastolic dysfunction [1]. Diastolic HF has been found to occur in more than 50% of patients with systolic HF [2,3]. The diastolic phase becomes shorter, which exacerbates the pre-existing impairment of left ventricular (LV) filling. Thus, diastolic irregularities lead to elevated pressure in the pulmonary circulation, causing shortness of breath [4]. Diastolic dysfunction is usually accompanying systolic dysfunction. Echocardiography is a key imaging method for the evaluation of diastolic function. Echocardiographic estimation of LV filling pressure can be drawn from algorithms accounting for Doppler velocities at the mitral valve, tissue Doppler imaging techniques and data of left atrium size [5,6].

There is a need for echocardiographic evaluation in all patients with HF in the qualification process for cardiac rehabilitation.

The most typical clinical symptoms reported by the patients are dyspnea and low exercise tolerance (fatigue and weakness upon exertion). Exercise dyspnea is also the earliest clinical manifestation in patients with diastolic HF, as tachycardia upon exertion triggers the pathomechanism of dyspnea. Thus, it is interesting to determine if cardiac rehabilitation can influence diastolic dysfunction in HF patients. HF is associated with progressive exercise intolerance. According to the 2020 Sports Cardiology ESC guidelines, exercise-based cardiac rehabilitation is recommended in all stable individuals with HF [7] to improve exercise capacity, quality of life, and to reduce the risk of the rehospitalization [8]. Because of the high mortality associated with chronic heart failure [9], there is need for wider implementation of evidence-based management.

The Telerehabilitation in Heart Failure Patients trial (TELEREH-HF) study [10,11] is the largest prospective, multicenter, and randomized clinical trial to date that assessed a 9-week hybrid comprehensive telerehabilitation (HCTR) compared to usual care (UC) in HF patients, and had the data regarding diastolic dysfunction in HF with reduced ejection fraction.

The TELEREH-HF trial supported the statement that telemedicine may offer a novel model of organization and HCTR may facilitate the implementation of the comprehensive management of HF patients. TELEREH proved that telerehabilitation is well accepted, safe, and effective with high adherence in HF patients. Our trial confirmed that HCTR improved quality of life in HF patients.

There are scarce data regarding the prognostic impact of diastolic dysfunction in HF patients, participating or not in cardiac telerehabilitation. What is more, HCTR is an attractive option for HF treatment during the COVID-19 pandemic.

### Research Objectives

The objective of the study was to evaluate the effects of individually prescribed HCTR on left ventricular diastolic function in HF patients. We focused on the impact of HCTR regarding the severity of diastolic dysfunction, mild versus severe. We assessed the survival probability depending on discrepancies in left ventricular diastolic function.

## 2. Methods

TELEREH-HF is a multicenter, prospective, randomized (1:1), open-label, parallel-group, controlled trial involving patients with HF assigned either to the HCTR program in conjunction with UC (HCTR group) or UC only (UC group). Patients were qualified for the TELEREH-HF study (Clinical Trials.gov NCT 02523560) with New York Heart Association (NYHA) class I, II or III HF with LV ejection fraction (LVEF) 40% or less after hospitalization due to worsening of HF within 6 months prior to randomization. The aim of the study was to determine whether the potential improvement in functional outcomes and quality of life after a 9-week training period improves clinical outcomes during an extended follow-up of 12 to 24 months.

The study was performed in accordance with the Helsinki Declaration and Polish legal regulations. Each patient gave informed consent. The study was approved by the Bioethics Committee at the National Institute of Cardiology in Warsaw. Patient data were verified by an independent Data Security Monitoring Council. The task of the Clinical Endpoint Committee, without knowledge of randomization, was to review hospitalizations and deaths.

The inclusion and exclusion criteria are presented elsewhere in design documents [10,11]. In the presented subanalysis, only patients with sinus rhythm were qualified. A patient in the HCTR group underwent a 9-week HCTR program with two stages: the first stage (1 week) was conducted in the hospital and the next stage (8 weeks) of HCTR was conducted at home 5 times a week. The telerehabilitation program includes three training ranges: aerobic endurance training, Nordic walking, breathing muscles training, exercises with light resistance and strength exercises.

### 2.1. Echocardiography Assessment

Echocardiography exams were performed by experienced echocardiographists on different echo machines on each site (GE Vivid 6, GE Vivid 4, Philips Epiq 8, Acuson CV70). Diameters of heart chambers were measured on long axis view, while left atrium volume was assessed in four-chamber apical view. The LVEF was determined by biplane Simpson’s method. Mitral inflow was evaluated by PW Doppler sample volume between mitral leaflet tips.

Mitral inflow was assessed by measurement of: early diastolic mitral inflow velocity (E wave), late diastolic mitral inflow velocity (A wave), deceleration time of E wave (DTE), and E/A ratio. On pulsed-wave tissue Doppler imaging, annular E’ velocity was measured on medial wall (E’ med.) and lateral wall of left ventricle (E’ lat). E velocity divided by mitral annular E’ velocity was calculated at medial wall (E/E’ med) and lateral wall (E/E’ lat), and then average value was calculated (E/E’ avg). Jet velocity of tricuspid regurgitation (TR) was calculated on continues wave Doppler. Normal mitral inflow was determined as both E/A ≤ 0.8 and E ≤ 50 cm/s. When mitral inflow shows an E/A ≤ 0.8 but the peak E velocity is >50 cm/sec, or if the E/A ratio is >0.8 but <2, other signals are necessary for accurate evaluation. Due to the lack of measurement of left atrium volume, we used only 2 criteria: TR jet peak velocity by color Doppler and average E/E’ ratio.

To determine diastolic dysfunction, we used algorithm for estimation of LV filling pressures and grading LV diastolic function in patients with HFrEF recommended by the American Society of Echocardiography and the European Association of Cardiovascular Imaging [5]. We excluded patients with AF because of differences of assessment of diastolic function in case of AF (altered pattern of mitral inflow, lack of A wave, variability in cycle length, common occurrence of LA enlargement regardless of filling pressures). I grade diastolic dysfunction with normal left atrium pressure was called as mild diastolic dysfunction (MDD). Severe diastolic dysfunction was defined as characterized by increased left atrium pressure, and so it consisted of both grade II and III dysfunction.

Thus, regarding grade of diastolic dysfunction, patients assigned to HCTR group were divided into HCTR-MDD and HCTR-SDD. Analogically, among patients from UC care, there were UC-MDD and UC-SDD subgroups.

Patients were followed during 14–26 months after for all-cause mortality, cardiovascular (CV) mortality, all-cause mortality, CV hospitalizations, HF hospitalizations, and composite points previously listed.

### 2.2. Statistical Analyses

Results are reported as numbers and percentages for categorical variables, or means ± SD (baseline characteristic) or means and 95% confidence intervals (difference between 9-week value and baseline) for continuous variables. Comparisons between groups on baseline characteristics were performed by the chi-square test of independence or the Fisher exact test (when the number of expected events was less than 5), the Cochran Mantel-Haenszel test, or Student’s *t*-test (or Satterthwaite method), respectively. Differences in change over time between groups were compared using a correction of variance analysis for baseline measurement level and body surface area, hypertension, loop diuretics, and NYHA class. Interactions between groups and diastolic dysfunction were studied. The rate of events (all-cause mortality, all-cause hospitalization, cardiovascular mortality, and cardiovascular hospitalization) was estimated using Kaplan–Meier curves and made using the log-rank test with the Tukey–Kramer correction for multiple comparisons. Two-sided *p* < 0.05 was considered statistically significant. The analyses were made using SAS statistical software version 9.4 (SAS Institute, Inc., Cary, NC, USA).

## 3. Results

Between the beginning of June 2015 and the end of June 2017, we randomized 850 eligible patients in a 1:1 ratio to either a HCTR plus usual care group (HCTR group) or a usual care only (UC group).

Among enrolled patients, sinus rhythm necessary for proper assessment of diastolic function was present in 512 patients. Echocardiography was performed twice before and after intervention (HCTR group) or observation (UC group) in 472 patients. The study flow diagram is shown in Figure 1.

Normal mitral inflow was found in 329 patients. First grade and mild diastolic dysfunction was found in 306 patients. Second grade diastolic dysfunction was present in 14 patients, when the restrictive pattern of mitral inflow with E/A ≥ 2 in 119 patients. Severe diastolic dysfunction was diagnosed in 119 patients. It was impossible to determine left atrial pressure and diastolic dysfunction in 28 patients.

Among patients assigned to the HCTR group with sinus rhythm, 168 patients had mild diastolic dysfunction (HCTR-MDD) and 67 patients had severe diastolic dysfunction (HCTR-SDD). Among patients assigned to the UC group with sinus rhythm, 143 patients had mild diastolic dysfunction (UC-MDD) and 66 patients had severe diastolic dysfunction (UC-SDD). On Figure 1, diastolic dysfunction criteria and groups are presented.

The study groups HCTR and UC did not significantly differ in terms of baseline clinical parameters, demographic data, and treatment, except for a higher prevalence of hypertension and more frequent use of loop diuretics in UC-SDD than in HCTR-SDD. Moreover, patients in the UC-MDD and HCTR-MDD groups differed in NYHA classes and body surface area. The baseline characteristics of the cohort at randomization are presented in Table 1. Echocardiographic parameters at randomization are listed in Table 2. There were only differences between HCTR-MDD and UC-MDD in DTE parameters at baseline.

Changes from baseline to 9 weeks in echocardiographic parameters were seen only in A velocities (delta from baseline to 9 weeks 0.06 (0.01;0.11) in HCTR-MDD vs. −0.03 (−0.11;0.04) in UC-MDD; *p* interaction = 0.008) and tricuspid regurgitation velocity (−0.10 (−0.28;0.08) in HCTR-SDD vs. 0.23 (−0.03;0.49) in UC-SDD; *p* interaction = 0.007). No significant shifts between groups of different diastolic dysfunction grade were observed after HCTR (Table 3). 

Understanding the impact of volume overload in HF patients, we were checking the weight gain before every session of cardiac rehabilitation. We did not notice any BMI changes of statistical importance between analyzed subgroups regarding diastolic dysfunction.

All-cause mortality was higher in UC-SDD vs. UC-MDD (24 (36.4%) vs. 42 (29.4%), *p* < 0.001), with no difference between HCTR-SDD and HCTR-MDD (24 (35.8%) vs. 49 (29.2%), *p* = 0.064) (Figure 2). No difference in the probability of CV mortality and hospitalization were found in HCTR and UC groups (Figure 3 and Figure 4). The probability of CV hospitalization was not associated with diastolic dysfunction. A higher probability of HF hospitalization (Figure 5) was seen in HCTR-SDD compared to HCTR-MDD (46 (68.6%) vs. 65 (38.7%), *p* < 0.001, retrospectively) and in UC-SDD compared to UC-MDD (40 (60.6%) vs. 55 (38.5%), *p* < 0.001, retrospectively).

## 4. Discussion

Currently, many research centers conduct projects aimed at optimizing non-invasive therapy in order to prevent HF progression [12]. Telemedicine is one of the solutions dedicated to this group of patients, also with HF [13]. Until now, there have been no large available data on the diastolic dysfunction in patients with HFrEF who were randomized to HCTR or UC. Our study, for the first time, describes the impact of diastolic dysfunction severity on prognosis in HF patients, in the context of a telerehabilitation program.

Imaging tests provide many important information necessary in the diagnosis and prognosis of patients with HF. In a group of 31 patients after an acute cardiovascular event, we assessed the effect of rehabilitation on the functional remodeling of the LV. It has been observed that rehabilitation leads to a reverse functional remodeling of the LV and an improvement in functional reserve [14]. Cardoso et al. also investigated the mechanisms associated with myocardial reverse remodeling in patients with HF with reduced and preserved LVEF, but still detailed data regarding the impact of cardiac rehabilitation on parameters of cardiac magnetic resonance imaging, vasomotor endothelial function, cardiac sympathetic activity imaging and serum biomarker are not available [15].

An important reason for examining diastolic function in patients with decreased LVEF is the assessment of LV filling pressure. Diastolic dysfunction with increased LV filling pressure determines the prognosis [16].

Diastolic dysfunction of the LV results in decreased exercise tolerance, and is associated with poor prognosis in patients, particularly the elderly [17]. Physical activity may improve clinical outcomes in patients with HF, including patients with end-stage HF treated with implantable devices to assist the LV [18]. However, the influence of exercise training on the diastolic function of the LV in patients with cardiovascular diseases remains controversial [19]. 

There are only a few papers regarding the influence of cardiac rehabilitation on diastolic function, mostly in patients with chronic coronary syndromes or after myocardial infarction.

Wuthiwaropas et al. analyzed the influence of a 3-month-old rehabilitation on the hemodynamic parameters of the myocardium in patients with coronary artery disease. Out of 24 (96%) patients: 12 (50%) had an improvement in diastolic function, 2 (8%) had a normal diastolic function all the time, 9 (38%) remained at the same level, and one (4%) had a deterioration in diastolic function [20]. At this point, it is worth emphasizing that the second largest group of patients studied did not benefit—the diastolic function, despite rehabilitation, did not change. Those results correspond to ours, but in the HF population.

Lee et al. determined the impact of cardiac rehabilitation on diastolic function and prognosis in patients after a history of acute myocardial infarction. The parameters E/E’ >14, velocity e’ of the septum <7 cm/s, left atrial volume index (LAVI) > 34 mL/m^2^ and maximum TR velocity > 2.8 m/s were compared. In the group undergoing cardiac rehabilitation, an improvement in the examined parameters was observed. The authors proved that cardiac rehabilitation was significantly associated with favorable diastolic function after myocardial infarction. Those results are in contrast to our study in the HF population, but in that study authors compared patients participating in cardiac rehabilitation sufficiently with those not participating sufficiently [21].

In the next study, 98 patients with moderate-to-severe, mild, and preserved LVEF were randomly assigned to exercise training plus UC or UC alone in a randomization ratio of 2:1. Cardiac rehabilitation increased the mean ratio of early-to-late mitral inflow velocities (E/A ratio) and decreased deceleration time (DT) of early filling in patients with mild and preserved LVEF. In patients with advanced diastolic dysfunction (DT < 160 ms), rehabilitation decreased E/A ratio and increased DT, both of which were unchanged after UC alone. Importantly, cardiac rehabilitation decreased left ventricular dimensions in patients with mild and moderate-to-severe reductions in LVEF but not in patients with preserved LVEF [22].

Pearson et al. evaluated the effect of exercise training on diastolic function in patients with HF. Data from five studies in HF with preserved ejection fraction (HFpEF) patients, with a total of 204 participants, also demonstrated a significant improvement in E/E’ in exercise group [23]. 

In the recent study on patients with acute coronary syndromes, the adopted criteria and detailed analysis of the tested diastolic dysfunction parameters did not show a significant effect of cardiac rehabilitation on diastolic function in the studied group. At this point, it is worth emphasizing that most of the patients enrolled in the study underwent STEMI. Moreover, the majority of respondents had a history of several years of high blood pressure. The matrix and the collagen fibers in the heart determine the effectiveness of the mechanical systole and diastole. Cardiac perfusion disorders activate macrophages and increase the concentration of transforming factors, e.g., TGF-beta1 (transforming growth factor beta 1). As a result, there is proliferation of fibroblasts and an increase in collagen content in the cell stroma and around the vessels [24]. STEMI is dominated by the process of structural degradation of collagen fibers under the influence of activated proteolytic enzymes. This starts stromal fibrosis with a disturbed ratio of collagen fibers, which increases muscle stiffness and generates disorders of its relaxation, and finally compliance. According to Soholm et al. Diastolic dysfunction in the early phase after STEMI determines the extent of myocardial damage and significantly reduces the effect of myocardial salvage treatment after three months. Thus, the presence of post-STEMI diastolic dysfunction is indicative of a poorer prognosis [25]. It is possible that the changes described in the study groups overlap with early changes generated by long-term hypertension, myocardial fibrosis and existing LV filling abnormalities. Therefore, they could observe permanent diastolic disorders, which, due to the irreversible nature of changes in the stroma, we are unable to reverse. Moreover, it is highly prevalent in hypertensive patients and is associated with increased cardiovascular morbidity and mortality [26]. Those results are not consistent with our results, because our population was HF patients.

The analysis carried out by Acar RD et al. was aimed at assessing the influence of cardiac rehabilitation on the LV diastolic function. The study was performed in a group of 82 patients after acute myocardial infarction. A significant improvement in the E/A wave ratio was observed; however, DTE and isovolumic relaxation time (IVRT) did not change significantly [27].

On the other hand, in another study, after an eight-week rehabilitation program in the group of patients after myocardial infarction, the authors also did not find a significant improvement in the examined echocardiographic parameters [28]. Similarly, after an 8-week endurance exercise program, despite the improvement in exercise capacity parameters, they did not notice a significant improvement in diastolic or systolic function [29]. The similarity to our work is the typical duration of cardiac rehabilitation.

Our study is the first randomized trial investigating the effect of comprehensive cardiac rehabilitation in HF patients with reduced ejection fraction on diastolic function. Some observed changes in echocardiographic parameters of diastolic function were not pronounced enough after 8-week HCTR. With the intention to see more clear beneficial effects of hybrid comprehensive telerehabilitation versus usual care rehabilitation on the diastolic function, the duration and volume of HCTR might be greater. We noticed an interesting observation regarding the prognostic impact of diastolic dysfunction severity on all-cause mortality in UC patients. Moreover, we observed a higher probability of HF hospitalization in case of SDD in both HCTR and UC arms.

## 5. Conclusions

Hybrid comprehensive telerehabilitation did not influence diastolic function in HF patients in a significant manner. The grade of diastolic dysfunction had an impact on mortality only in the UC group and HF hospitalization over a 12–24-month follow-up in HCTR and UC groups. Nevertheless, it is well known that cardiac rehabilitation in patients with HF may reduce the risk of rehospitalization and may reduce HF-related hospital admissions. The use of modern technologies for HCTR is helpful to overcome accessibility barriers to cardiac rehabilitation. HCTR should be considered a tool of great importance in HF patients.

### Limitations

Our conclusions are drawn up only in patients with sinus rhythm, when atrial fibrillation is not uncommon in HF patients. In our study, the lack of influence of HCTR on diastolic function can be explained by its duration of 8 weeks. To see a better effect, longer probably and a more intensive program are needed. We could not determine the grade of diastolic dysfunction in some patients because of the lack of biplane measurement of left atrium volume. According to the recent echocardiographic guidelines, we used two criteria during the second step of diastolic function assessment after the characterization of mitral inflow.

## Figures and Tables

**Figure 1 jcm-11-01844-f001:**
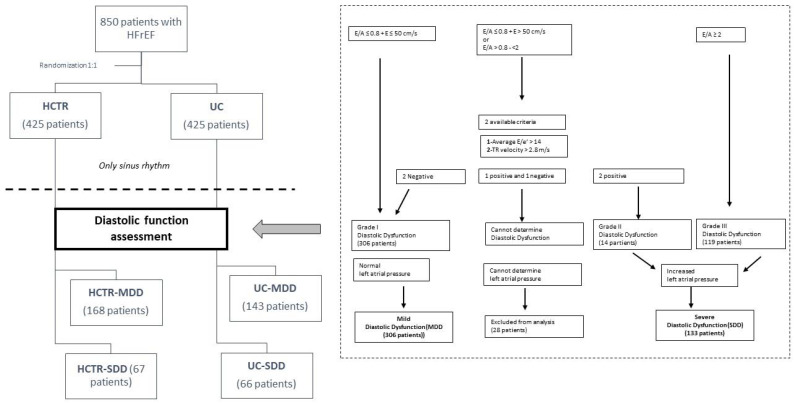
Study flow with algorithm for estimation of LV filling pressures and grading LV diastolic function in patients with HFrEF.

**Figure 2 jcm-11-01844-f002:**
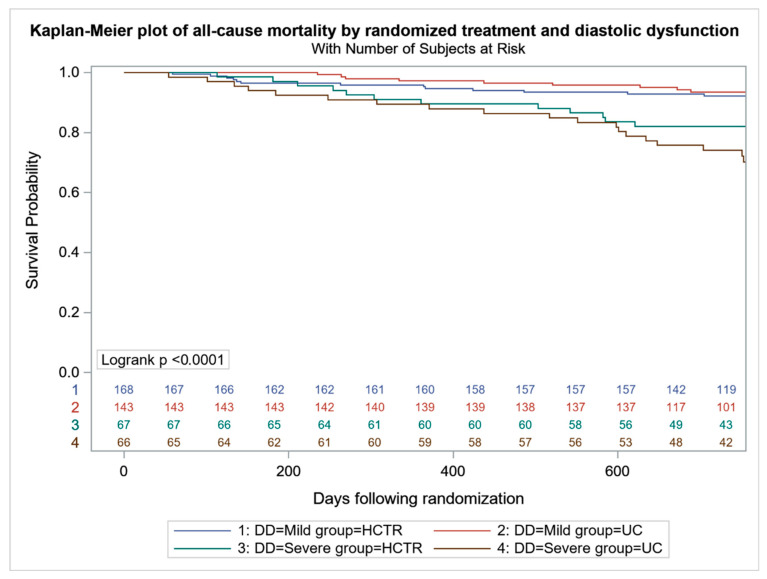
Kaplan–Meier Plot of all-cause mortality-free survivals in subgroups regarding diastolic function and rehabilitation.

**Figure 3 jcm-11-01844-f003:**
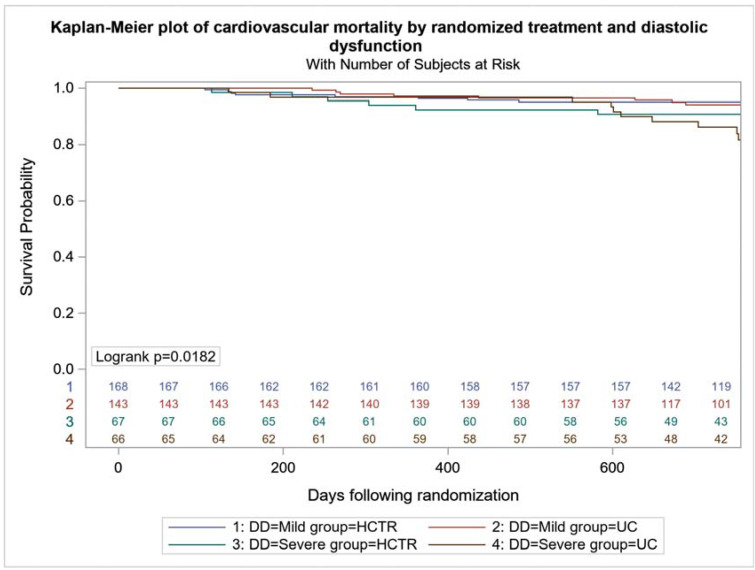
Kaplan–Meier plot of cardiovascular mortality-free survival in subgroups regarding diastolic function and rehabilitation.

**Figure 4 jcm-11-01844-f004:**
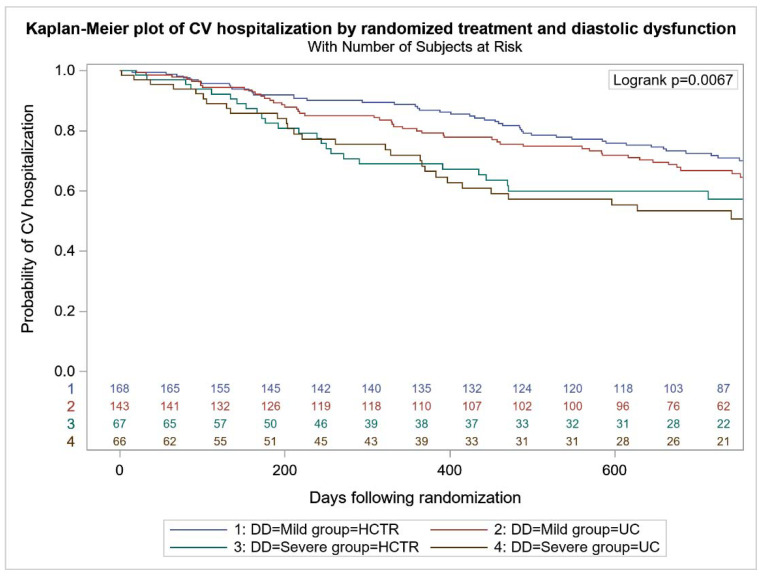
Kaplan–Meier plot of cardiovascular hospitalization in subgroups regarding diastolic function and rehabilitation.

**Figure 5 jcm-11-01844-f005:**
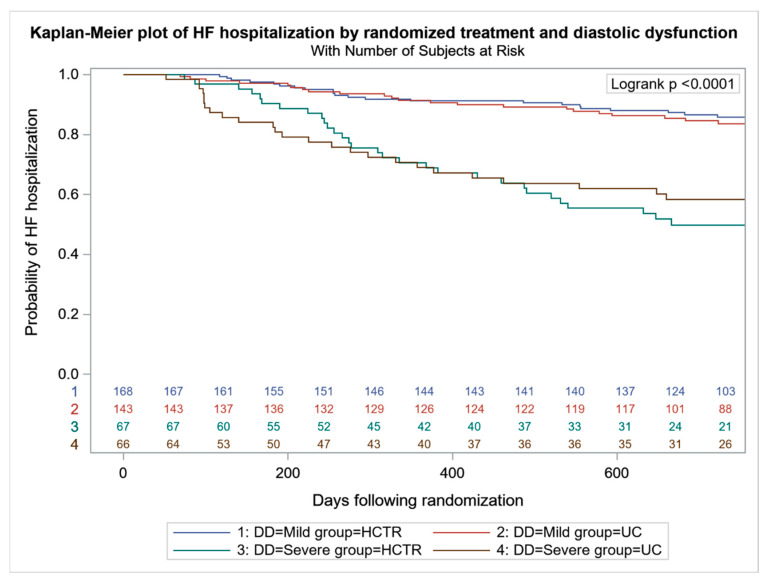
Kaplan–Meier plot of heart failure hospitalization in subgroups regarding diastolic function and rehabilitation.

**Table 1 jcm-11-01844-t001:** Baseline characteristics.

	MDD (*n* = 311)			SDD (*n* = 133)		
	HCTR-MDD *n* = 168	UC-MDD *n* = 143	*p*1	HCTR-SDD *n* = 67	UC-SDD *n* = 66	*p*2
Males. *n* (%)	151 (89.9)	131 (91.6)	0.602	59 (88.1)	57 (86.4)	0.770
Age (years). mean ± SD	60.9 ± 10.8	60.9 ± 10.3	0.977	62.3 ± 13.6	62.6 ± 10.2	0.911
Left Ventricular Ejection Fraction (%). mean ± SD	32.7 ± 6.2	32.2 ± 6.7	0.552	27.8 ± 6.3	27.7 ± 7.4	0.931
BSA (m^2^)	2.01 ± 0.22	2.06 ± 0.21	0.038	1.93 ± 0.20	1.96 ± 0.20	0.419
Etiology of heart failure. *n* (%)						
Ischaemic	117 (69.6)	90 (62.9)	0.212	43 (64.2)	45 (68.2)	0.626
Non-ischeamic	51 (30.4)	53 (37.1)		24 (35.8)	21 (31.8)	
Previous medical history. *n* (%)						
Coronary artery disease	115 (65.4)	88 (61.5)	0.202	44 (65.7)	45 (68.2)	0.758
Myocardial infarction	104 (61.9)	81 (56.6)	0.346	44 (65.7)	39 (59.1)	0.433
Angioplasty	79 (47.0)	66 (46.1)	0.878	33 (49.2)	36 (54.5)	0.541
Coronary artery bypass grafting	25 (14.9)	21 (11.7)	0.961	11 (16.4)	8 (12.1)	0.480
Hypertension	101 (60.1)	97 (67.8)	0.159	34 (50.7)	45 (68.2)	0.041
Stroke	9 (5.4)	7 (4.9)	0.854	2 (3.0)	8 (12.1)	0.055
Chronic kidney disease	21 (12.5)	19 (13.3)	0.836	18 (26.9)	14 (21.2)	0.446
Hyperlipidemia	85 (50.6)	63 (44.1)	0.250	31 (46.3)	27 (40.9)	0.533
Diabetes	56 (33.3)	47 (32.9)	0.931	21 (31.3)	24 (36.4)	0.541
Functional status						
NYHA I. *n* (%)	19 (11.3)	32 (22.4)	0.007	8 (11.9)	3 (4.5)	0.254
NYHA II. *n* (%)	127 (75.6)	85 (59.4)		45 (67.2)	45 (68.2)	
NYHA III. *n* (%)	22 (13.1)	26 (18.2)		14 (20.9)	18 (27.3)	
Treatment						
Beta-blocker	161 (95.8)	137 (95.8)	0.990	63 (94.0)	66 (100)	0.119
ACEI/ARB	159 (94.6)	137 (95.8)	0.634	58 (86.6)	58 (87.9)	0.821
Digoxin	9 (5.4)	5 (3.5)	0.430	8 (11.9)	5 (7.6)	0.397
Loop diuretics	116 (69.0)	100 (69.9)	0.866	52 (77.6)	61 (92.4)	0.017
Spironolactone/eplerenone	138 (82.1)	118 (82.5)	0.931	54 (80.6)	53 (80.3)	0.966
Aspirin/clopidogrel	121 (72.0)	89 (62.2)	0.066	37 (55.2)	43 (65.1)	0.243
Anticoagulants	23 (13.7)	22 (15.4)	0.672	19 (28.4)	18 (27.3)	0.889
Statins	146 (86.9)	120 (83.9)	0.455	50 (74.6)	52 (78.8)	0.570
CIEDs	122 (72.6)	117 (81.8)	0.055	58 (86.6)	54 (81.8)	0.453
Implantable cardioverter-defibrillator	75 (61.5)	78 (66.7)	0.482	39 (67.2)	33 (61.1)	0.310
CRT-P	3 (2.5)	2 (1.7)		0	0	
CRT-D	42 (34.4)	37 (31.6)		19 (32.8)	19 (35.2)	

Abbreviations: NYHA—New York Heart Association class; ACEI—angiotensin-converting enzyme inhibitors; ARB—angiotensin receptor blockers; CIEDs—cardiovascular implantable electronic devices; CRT-P—cardiac resynchronization therapy; CRT-D—cardiac resynchronization therapy and cardioverter-defibrillator; DM—diabetes mellitus; BSA—body surface area; HCTR-MD—patients in hybrid comprehensive telerehabilitation arm with mild diastolic dysfunction; HCTR-SDD—patients in hybrid comprehensive telerehabilitation arm with severe diastolic dysfunction; HFrEF—heart failure with reduced ejection fraction; UC-MDD—patients in usual care arm with mild diastolic dysfunction; UC-SDD—patients in usual care arm with severe diastolic dysfunction.

**Table 2 jcm-11-01844-t002:** Baseline parameters of echocardiographic parameters.

	MDD (*n* = 311)			SDD (*n* = 133)		
	HCTR-MDD *n* = 168	UC-MDD *n* = 143	*p*1	HCTR-SDD *n* = 67	UC-SDD *n* = 66	*p*2
E	0.53 ± 0.16	0.55 ± 0.17	0.421	0.93 ± 0.24	0.93 ± 0.24	0.962
A	0.70 ± 0.17	0.73 ± 0.17	0.219	0.39 ± 0.18	0.43 ± 0.18	0.224
E/A	0.79 ± 0.29	0.79 ± 0.33	0.983	2.63 ± 0.91	2.43 ± 0.97	0.267
DTE	230 ± 59	222 ± 64	0.241	164 ± 49	172 ± 54	0.432
E/E’ lat	8.09 ± 3.07	8.03 ± 3.02	0.883	17.3 ± 8.6	15.4 ± 7.1	0.223
E/E’ med	9.53 ± 3.2	9.50 ± 3.07	0.935	19.1 ± 10.0	22.5 ± 10.1	0.067
E/E’ avg	8.81 ± 2.61	8.77 ± 2.46	0.904	18.2 ± 8.3	19.0 ± 7.6	0.586
LA	42.8 ± 6.0	43.2 ± 5.8	0.546	46.3 ± 7.0	47.6 ± 6.4	0.254
LAA	23.0 ± 4.8	24.0 ± 5.9	0.113	28.5 ± 6.2	28.2 ± 6.8	0.854
TR velocity	2.03 ± 0.46	2.04 ± 0.49	0.894	2.8 ± 0.7	2.9 ± 0.7	0.557
EF	32.8 ± 6.1	32.1 ± 6.7	0.330	27.9 ± 6.4	28.0 ± 7.4	0.937
E/A ≤ 0.8 (*n*, %)	112 (66.7)	98 (68.5)	0.727	1 (1.5)	1 (1.5)	0.185
E/A 0.8–2 (*n*, %)	56 (33.3)	45 (31.5)		3 (4.5)	9 (13.6)	
E/A > 2 (*n*, %)	0	0		63 (94.0)	56 (84.8)	
DTE ≤ 160 (*n*, %)	15 (9.0)	18 (12.6)	0.004	37 (56.1)	34 (53.1)	0.870
DTE 160–200 (*n*, %)	26 (15.7)	10 (29.4)		17 (25.8)	16 (25.0)	
DTE ≥ 200 (*n*, %)	125 (75.3)	83 (58.0)		12 (18.2)	25 (21.9)	
E/E’ avg > 14 (*n*, %)	1 (0.6)	2 (1.4)	0.595	44 (65.7)	49 (75.4)	0.221
TR velocity > 2.8 (*n*, %)	1 (0.6) (*n* = 141)	0 (0) (*n* = 119)	1.00	23 (43.4) (*n* = 52)	27 (51.9) (*n* = 53)	0.382

Abbreviations: HCTR-MDD—patients in hybrid comprehensive telerehabilitation arm with mild diastolic dysfunction; HCTR-SDD– patients in hybrid comprehensive telerehabilitation arm with severe diastolic dysfunction; HFrEF—heart failure with reduced ejection fraction; UC-MDD—patients in usual care arm with mild diastolic dysfunction; UC-SDD—patients in usual care arm with severe diastolic dysfunction; LVEF—Left Ventricular Ejection Fraction; E—early diastolic mitral inflow velocity; A—late diastolic mitral inflow velocity; DTE—deceleration time of E wave; E’ med—E’ velocity at medial wall; E’ lat—E’ velocity at lateral wall; E/E’ avg—average value of E/E’ at medial and lateral wall of the left ventricle; TR velocity—tricuspid regurgitation (TR) jet velocity; LA—left atrium diameter; LAA—left atrium area.

**Table 3 jcm-11-01844-t003:** Changes from baseline to 9 weeks in echocardiographic parameters (adjusted for baseline measure, body Surface area, hypertension, loop diuretics, NYHA class).

	MDD (*n* = 311)				SDD (*n* = 133)				
	HCTR-MDD *n* = 168	UC-MDD *n* = 143	Difference [95% CI] *	*p* *	HCTR-SDD *n* = 67	UC-SDD *n* = 66	Difference [95% CI] *	*p* *	*p* Interaction
	Δ 9 week—baseline [95% CI] *				Δ 9 week—baseline [95% CI] *				
E [m/s]	0.01 (−0.03;0.08)	−0.02 (−0.05;0.01)	0.03 (−0.02;0.08)	0.526	0.01 (−0.05;0.05)	0.03 (−0.02;0.08)	−0.02 (−0.10;−0.05)	0.864	0.163
A	0.03 (−0.01;0.06)	−0.03 (−0.06;−0.01)	0.06 (0.01;0.11)	<0.010	−0.01 (−0.06;0.04)	0.02 (−0.02;0.07)	−0.03 (−0.11;0.04)	0.648	0.008
E/A	−0.03 (−0.14;0.09)	−0.02 (−0.13;0.10)	−0.01 (−0.19;0.17)	0.992	−0.03 (−0.24;0.19)	0.00 (−0.20;0.21)	−0.03 (−0.32;0.25)	0.989	0.844
DTE	7.7 (−3.9;19.4)	4.5 (−6.7;15.8)	3.2 (−16.0;22.4)	0.973	−23.9 (−41.6;−6.1)	−22.6 (−39.9;−5.3)	−1.3 (−31.1;28.6)	0.999	0.744
E/E’ lat	−0.08 (−1.05;0.90)	−0.27 (−1.21;0.67)	0.19 (−1.37;1.76)	0.989	2.07 (0.53;3.61)	1.42 (−0.02;2.86)	0.65 (−1.78;3.61)	0.901	0.684
E/E’ med	−0.78 (−1.92;0.35)	−0.97 (−2.07;0.13)	0.19 (−1.63;2.01)	0.993	1.86 (0.14;3.58)	2.38 (0.56;4.20)	−0.52 (−3.36;2.32)	0.965	0.587
E/E’ avg	−0.32 (−1.25;0.61)	−0.51 (−1.41;0.40)	0.19 (−1.29;1.66)	0.988	1.79 (0.34;3.24)	1.55 (0.09;3.01)	0.24 (−2.04;2.52)	0.993	0.958
LA LAX	0.03 (−0.83;0.89)	−0.66 (−1.49;0.17)	0.69 (−0.73;2.11)	0.597	0.72 (−0.55;1.98)	−0.24 (−1.50;1.02)	0.96 (−1.22;3.13)	0.670	0.789
LAA 4CH	−0.24 (−0.98;0.49)	−0.69 (−1.40;0.01)	0.45 (−0.75;1.66)	0.764	1.70 (0.55;2.85)	1.14 (0.05;2.23)	0.56 (−1.35;2.46)	0.875	0.904
TR velocity	−0.17 (−0.28;−0.06)	−0.07 (−0.18;0.04)	−0.10 (−0.28;0.08)	0.462	0.18 (0.02;0.34)	−0.05 (−0.21;0.10)	0.23 (−0.03;0.49)	0.106	0.007
EF	2.17 (1.45;2.88)	1.14 (0.45;1.82)	1.03 (−0.15;2.21)	0.111	1.06 (0.01;2.11)	1.27 (0.23;2.31)	−0.21 (−2.03;1.60)	0.990	0.138

Abbreviations: HCTR-MDD—patients in hybrid comprehensive telerehabilitation arm with mild diastolic dysfunction; HCTR-SDD–patients in hybrid comprehensive telerehabilitation arm with severe diastolic dysfunction; HFrEF—heart failure with reduced ejection fraction; UC-MDD—patients in usual care arm with mild diastolic dysfunction; UC-SDD—patients in usual care arm with severe diastolic dysfunction; LVEF—left ventricular ejection fraction; E—early diastolic mitral inflow velocity; A—late diastolic mitral inflow velocity; DTE—deceleration time of E wave; E’ med—E’ velocity at medial wall; E’ lat—E’ velocity at lateral wall; E/E’ avg—average value of E/E’ at medial and lateral wall of the left ventricle; TR velocity—tricuspid regurgitation (TR) jet velocity; LA—left atrium diameter; LAA—left atrium area. * regarding Difference [95% CI].

## Data Availability

The data used to support the findings of this study are included within the article.

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
