# Peer review of "Prognostic Impact of Hybrid Comprehensive Telerehabilitation Regarding Diastolic Dysfunction in Patients with Heart Failure with Reduced Ejection Fraction—Subanalysis of the TELEREH-HF Randomized Clinical Trial"

_jcm, 2022, doi:10.3390/jcm11071844_

Round 1

Reviewer 1 Report

This is a quite interesting paper on the a hybrid tele-rehabilitation in patients with heart failure.

Just have a few comments:

  1. Please spell out the full name for an abbreviations in its first appearance in Abstract and Text, so that readers are more likely to follow.
  2. Line 86. The sub-analysis only included those with sinus rhythm. What is the rational for this? Are there any difference in characteristics for those included and not included?
  3. Line 126. The authors used mean and SD to describe continuous variables. So, are all continuous variables followed normal distribution? 

Author Response

Dear Reviewer,

Thank you for the discerning comments concerning our manuscript entitled “Prognostic impact of hybrid comprehensive telerehabilitation regarding diastolic dysfunction in patients with heart failure with reduced ejection fraction - subanalysis of the TELEREH-HF randomized clinical trial” which we accept with comprehension and gratitude. We have studied your comments carefully and made corrections which we hope will meet with your approval. Your questions or comments are answered in detail below, with original reviewer comments denoted in boldface, our responses in regular typeface and all changes in the manuscript in red and bold font.

Due to remark regarding repetition, we modified some paragraphs, especially Methods and Statistical analyses sections. Because of specific terminology of echocardiography, some terms might be repeated. We tried to paraphrase them all and marked by red font.

Detailed answers to review comments:

This is a quite interesting paper on the a hybrid tele-rehabilitation in patients with heart failure.

Just have a few comments:

  1. Please spell out the full name for an abbreviations in its first appearance in Abstract and Text, so that readers are more likely to follow.

Response: We agree.

Change: We provided explanation for all abbreviations. For example in Abstract:

The objective of the study was to evaluate the effects of individually prescribed hybrid comprehensive telerehabilitation (HCTR) implemented at patients’ homes on left ventricular (LV) diastolic function  in heart failure (HF) patients.

  1. Line 86. The sub-analysis only included those with sinus rhythm. What is the rational for this? Are there any difference in characteristics for those included and not included?

Response: As we mentioned: ‘sinus rhythm necessary for proper assessment of diastolic function’. We analysis diastolic dysfunction using algorithm for estimation of LV filling pressures and grading LV diastolic function in patients with HFrEF recommended by the American Society of Echocardiography and the European Association of Cardiovascular Imaging. According to those guidelines, the approach to evaluate diastolic function starts with mitral inflow velocities and is applied in the absence of atrial fibrillation (AF), because in in case og AF there is lack of A wave of mitral inflow. Thus there is impossible to calculate E/A ratio. What is more, in patients with AF, Doppler assessment of LV diastolic function is limited by the variability in cycle length, the absence of organized atrial activity, and the frequent occurrence of LA enlargement regardless of filling pressures. There is discussion about limitation of prognostic implication of assessment of diastolkic dysfunction in patents with atrial fibrillation versus sinus rhythm

Change: We added sentence in ‘Echocardiography assessment’:

We excluded patients with AF because of differences of assessment of diastolic function in case of AF (altered pattern of mitral inflow, lack of A wave, , variability in cycle length, common  occurrence of LA enlargement regardless of filling pressures).

  1. Line 126. The authors used mean and SD to describe continuous variables. So, are all continuous variables followed normal distribution? 

Response: Distribution of some variables in dome subgroups deviates from the normal distribution, but samples were sufficiently large (>50)  to justify their use based on the Central Limit Theorem.

Reviewer 2 Report

Dear authors, 

it is a topic of great interest in cardiology and any information resulting from the research helps us all in developing this field. 

However, some revisions are needed. 

The introductory chapter is very brief. It should be detailed with more information about heart failure and the current state of knowledge in terms of epidemiology, pathophysiology and echocardiography. 

I would also suggest to add a short subchapter with a briefly presentation of TELEREH-HF purpose and results.  

The last subchapter of the introduction should be the “results objectives”. The recommendation here would be to introduce the purpose of the subanalysis, the expected results and/or impact. 

The chapters of methods, results and discussion are presented in detail and understandably, so I have no recommendation of change in these sections.

The chapter of conclusions I consider should contain some personal ideas about how the results of this study and the large study impact the implementation of telerehabilitation globally. 

Kind regards.

Author Response

Dear Reviewer,

Thank you for the discerning comments concerning our manuscript entitled “Prognostic impact of hybrid comprehensive telerehabilitation regarding diastolic dysfunction in patients with heart failure with reduced ejection fraction - subanalysis of the TELEREH-HF randomized clinical trial” which we accept with comprehension and gratitude. We have studied your comments carefully and made corrections which we hope will meet with your approval. Your questions or comments are answered in detail below, with original reviewer comments denoted in boldface, our responses in regular typeface and all changes in the manuscript in red and bold font.

Due to remark regarding repetition, we modified some paragraphs, especially Methods and Statistical analyses sections. Because of specific terminology of echocardiography, some terms might be repeated. We tried to paraphrase them all and marked by red font.

Detailed answers to review comments:

it is a topic of great interest in cardiology and any information resulting from the research helps us all in developing this field. 

However, some revisions are needed. 

Response: We would like to thank you for very kind words in your review and we corrected the manuscript according to your advices.

The introductory chapter is very brief. It should be detailed with more information about heart failure and the current state of knowledge in terms of epidemiology, pathophysiology and echocardiography. 

Response: We agree.

Change: We rewrote the Introduction chapter and added new relevant references (1,6,7,8,9)

Heart failure (HF) is a major challenge in modern healthcare and is increasing with the aging of the population. The pathophysiology in HF is determined by altered cardiac output, reduced cardiac contractility, myocardial stiffness, increased filling pressure of LV and diastolic dysfunction1. Diastolic HF has been found to occur in more than 50% patients with systolic HF.2,3 The diastolic phase becomes shorter, which exacerbates the preexisting impairment of left ventricular (LV) filling. Thus, diastolic irregularities lead to elevated pressure in the pulmonary circulation, causing shortness of breath.4 Diastolic dysfunction is usually accompanying systolic dysfunction. Echocardiography is key imaging method for evaluation of diastolic function. Echocardiographic estimation of LV filling pressure can be drawn from algorithms accounting Doppler velocities at mitral valve, tissue Doppler imaging techniques and data of left atrium size5,6.

There is a need for echocardiographic evaluation in all patients with HF in qualification process for cardiac rehabilitation.

The most typical clinical symptoms reported by the patients are dyspnea and low exercise tolerance (fatigue and weakness upon exertion). Exercise dyspnea is also the earliest clinical manifestation in patients with diastolic HF as tachycardia upon exertion triggers the pathomechanism of dyspnea. Thus it is interesting if cardiac rehabilitation can influence on diastolic dysfunction in HF patients. HF is associated with progressive exercise intolerance. According to the 2020 Sports Cardiology ESC guidelines exercise-based cardiac rehabilitation is recommended in all stable individuals with HF7 to improve exercise capacity, quality of life, and to reduce risk of the rehospitalization8. Because of high mortality associated with chronic heart failure9, there is need for wider implementation of evidence-based management.

The Telerehabilitation in Heart Failure Patients trial (TELEREH-HF) study10,11 is the largest prospective, multicenter, and randomized clinical trial to date that assessed a 9-week hybrid comprehensive telerehabilitation (HCTR) compared to usual care (UC) in HF patients, and had the data regarding diastolic dysfunction in HF with reduced ejection fraction.

TELEREH-HF trial supported statement that telemedicine may offer a novel model of organization and HCTR may facilitate implementation of comprehensive management of HF patients. TELEREH proved that telerehabilitation is well accepted, safe, effective with high adherence in HF patients. Our trial confirmed that HCTR improved quality of life in HF patients.

There are scarce data regarding prognostic impact of diastolic dysfunction in HF patients, participating or not in cardiac telerehabilitation. What is more, HCTR is attractive option for HF treatment during the COVID-19 pandemic.

I would also suggest to add a short subchapter with a briefly presentation of TELEREH-HF purpose and results.  

Response: we agree.

Change: We added in Introduction chapter some sentences regarding main results of TELEREH-HF trial:

TELEREH-HF study is the largest prospective, multicenter, and randomized clinical trial to date that assessed a 9-week hybrid comprehensive telerehabilitation (HCTR) compared to usual care (UC) in HF patients, and had the data regarding diastolic dysfunction in HF with reduced ejection fraction.

TELEREH trial supported statement that telemedicine may offer a novel model of organization and HCTR may facilitate implementation of comprehensive management of HF patients. TELEREH proved that telerehabilitation is well accepted, safe, effective with high adherence in HF patients. Our trial confirmed that HCTR improved quality of life in HF patients.

The last subchapter of the introduction should be the “results objectives”. The recommendation here would be to introduce the purpose of the subanalysis, the expected results and/or impact. 

Response: Below Introduction chapter there is chapter with Research objectives. We decided to expand this chapter.

Change:

The objective of the study was to evaluate the effects of individually prescribed HCTR on left ventricular diastolic function in HF patients. We focused on impact of HCTR regarding severity of diastolic dysfunction, mild versus severe. We assessed the survival probability depending discrepancies in left ventricular diastolic function.

The chapters of methods, results and discussion are presented in detail and understandably, so I have no recommendation of change in these sections.

Response: We are grateful for this kind comment.

The chapter of conclusions I consider should contain some personal ideas about how the results of this study and the large study impact the implementation of telerehabilitation globally. 

Response: We agree.

Change: We addes 2 sentences regarding importance of HCTR in HF patients.

Hybrid comprehensive telerehabilitation did not influence on diastolic function in HF patients, in significant manner. Grade of diastolic dysfunction had impact  on mortality only in UC group and HF hospitalization over 12-24-month follow-up  in HCTR and UC groups .

Nevertheless, it is well known cardiac rehabilitation in patients with HF may reduce the risk of rehospitalization and may reduce HF-related hospital admissions. Use of modern technologies for HCTR is helpful to overcome accessibility barriers to cardiac rehabilitation. HCTR should be considered as a tool of great importance in HF patients.

Reviewer 3 Report

1) General comments

In their original manuscript entitled “Prognostic impact of hybrid comprehensive telerehabilitation regarding diastolic dysfunction in patients with heart failure with reduced ejection fraction – subanalysis of the TELEREH-HF randomized clinical trial”, the authors reported that hybrid comprehensive telerehabilitation did not influence on diastolic function in heart failure patients.  As they mentioned that this hybrid comprehensive telerehabilitation is attractive option during this COVID era.  This is an important report and I have a few minor comments.

2) Specific revision comments:

Major comments

  1. Could the authors report the trend of body weight and/or BMI for this study? We cannot miss this data in order to assess heart failure patients. 

Minor comments

  1. First abbreviation needs to be spelled out. For instance, “HCTR” (Page 1, Line 28) in abstract has to be written “Hybrid comprehensive telerehabilitation (HCTR).  “HCTR” should be used in the beginning of the “Conclusion” in abstract (Page 1, Line 40).
  2. Please mention that the study group was only focused on for sinus rhythm patients in the abstract.
  3. Figure 4 and 5: In terms of vertical axis, the label should not be survival probability.
  4. Page 10, Line 227: typo “available.s.”
  5. Please emphasize that the study was limited for sinus rhythm patients in the conclusion.

Author Response

Dear Reviewer,

Thank you for the discerning comments concerning our manuscript entitled “Prognostic impact of hybrid comprehensive telerehabilitation regarding diastolic dysfunction in patients with heart failure with reduced ejection fraction - subanalysis of the TELEREH-HF randomized clinical trial” which we accept with comprehension and gratitude. We have studied your comments carefully and made corrections which we hope will meet with your approval. Your questions or comments are answered in detail below, with original reviewer comments denoted in boldface, our responses in regular typeface and all changes in the manuscript in red and bold font.

Due to remark regarding repetition, we modified some paragraphs, especially Methods and Statistical analyses sections. Because of specific terminology of echocardiography, some terms might be repeated. We tried to paraphrase them all and marked by red font.

Detailed answers to review comments:

1) General comments

In their original manuscript entitled “Prognostic impact of hybrid comprehensive telerehabilitation regarding diastolic dysfunction in patients with heart failure with reduced ejection fraction – subanalysis of the TELEREH-HF randomized clinical trial”, the authors reported that hybrid comprehensive telerehabilitation did not influence on diastolic function in heart failure patients.  As they mentioned that this hybrid comprehensive telerehabilitation is attractive option during this COVID era.  This is an important report and I have a few minor comments.

2) Specific revision comments:

Major comments

  1. Could the authors report the trend of body weight and/or BMI for this study? We cannot miss this data in order to assess heart failure patients. 

Response:

Before beginning a training session, patients answer a series of questions regarding their present condition: fatigue, dyspnea, blood pressure, body mass, and medication taken. We could observed trends in patients’ body weight between rehabilitation sessions, but we did not notice any changes of statistical important between analyzed subgroups regarding diastolic dysfunction:

BMI baseline

BMI after 9 weeks

P

All patients

MDD

29.0 ± 4.7

29.3 ± 5.0

p=0.164

SDD

27.6 ± 5.0

27.8 ± 4.7

p=0.180

HCTR group

MDD

28.9 ± 4.9

29.2 ± 5.3

p=0.179

SDD

27.3 ± 5.0

27.5 ± 4.6       

p=0.284

UC group

MDD

29.3 ± 4.4

29.3 ± 4.6

p=0.669

SDD

27.9 ± 5.1

28.1 ± 4.9       

p=0.416

Change:

We added 2 sentences in Results section:

Understanding the impact of volume overload in HF patients, we were checking the weight gain before every session of cardiac rehabilitation. We did not notice any BMI changes of statistical importance between analyzed subgroups regarding diastolic dysfunction.

Minor comments

  1. First abbreviation needs to be spelled out. For instance, “HCTR” (Page 1, Line 28) in abstract has to be written “Hybrid comprehensive telerehabilitation (HCTR).  “HCTR” should be used in the beginning of the “Conclusion” in abstract (Page 1, Line 40).

Response: we agree and we corrected the manuscript according to your advices.

  1. Please mention that the study group was only focused on for sinus rhythm patients in the abstract.

Response: We agree.

Change: We added 1 sentence in the Abstract.

Due to difficultties of proper assessement and differences in evaluation of diastolic function in patients with atrial fibrillation, we included to our subanalysis only patients with sinus rhythm. Depends on grade of diastolic dysfunction patients assigned to subgroups with mild diastolic (MDD) or severe diastolic dysfunction (SDD), both in HCTR (HCTR-MDD and HCTR-SDD) and UC groups (UC-MDD and UC-SDD).

  1. Figure 4 and 5: In terms of vertical axis, the label should not be survival probability.

Response: We agree.

Change: We uploaded corrected figures with new annotations on figures:

Figure 4: 'Probability of  cardiovascular hospitalization'
Figure 5: 'Probability of heart failure hospitalization'

  1. Page 10, Line 227: typo “available.s.”

Response: We colrrected that  spelling mistake.

  1. Please emphasize that the study was limited for sinus rhythm patients in the conclusion.

Response: We agree that should be emphasize and we added that in Limitation chapter just after chapter with Conclusions.

Change: There is new sentence at the beginning of Limitation chapter:

Our conclusions are drawn up only in patients with sinus rhythm, when atrial fibrillation is not uncommon in HF patients.
